# New Insights into ADAMTS Metalloproteases in the Central Nervous System

**DOI:** 10.3390/biom10030403

**Published:** 2020-03-05

**Authors:** Yamina Mohamedi, Tania Fontanil, Teresa Cobo, Santiago Cal, Alvaro J. Obaya

**Affiliations:** 1Departamento de Bioquímica y Biología Molecular, Universidad de Oviedo, 33006 Oviedo, Asturias, Spain; yamomu@hotmail.com (Y.M.); taniuskina@gmail.com (T.F.); santical@uniovi.es (S.C.); 2Departamento de Biología Funcional, Área de Fisiología, Universidad de Oviedo, 33006 Oviedo, Asturias, Spain; 3Instituto Universitario de Oncología, IUOPA, Universidad de Oviedo, 33006 Oviedo, Asturias, Spain; 4Departamento de Investigación, Instituto Ordóñez, 33012 Oviedo, Asturias, Spain; 5Departamento de Cirugía y Especialidades Médico-Quirúrgicas, Universidad de Oviedo, 33006 Oviedo, Asturias, Spain; teresa@iaodontologia.es; 6Instituto Asturiano de Odontología, 33006 Oviedo, Asturias, Spain

**Keywords:** ADAMTS, extracellular matrix, central nervous system, proteoglycan, lectican, hyalectan

## Abstract

Components of the extracellular matrix (ECM) are key players in regulating cellular functions throughout the whole organism. In fact, ECM components not only participate in tissue organization but also contribute to processes such as cellular maintenance, proliferation, and migration, as well as to support for various signaling pathways. In the central nervous system (CNS), proteoglycans of the lectican family, such as versican, aggrecan, brevican, and neurocan, are important constituents of the ECM. In recent years, members of this family have been found to be involved in the maintenance of CNS homeostasis and to participate directly in processes such as the organization of perineural nets, the regulation of brain plasticity, CNS development, brain injury repair, axonal guidance, and even the altering of synaptic responses. ADAMTSs are a family of “A disintegrin and metalloproteinase with thrombospondin motifs” proteins that have been found to be involved in a multitude of processes through the degradation of lecticans and other proteoglycans. Recently, alterations in ADAMTS expression and activity have been found to be involved in neuronal disorders such as stroke, neurodegeneration, schizophrenia, and even Alzheimer’s disease, which in turn may suggest their potential use as therapeutic targets. Herein, we summarize the different roles of ADAMTSs in regulating CNS events through interactions and the degradation of ECM components (more specifically, the lectican family of proteoglycans).

## 1. Introduction

The extracellular matrix (ECM) constitutes a complex environment that is mainly formed by proteins and carbohydrates, where cells perform all required activities to control individual homeostasis [1]. In fact, cellular functions regulate the composition of the ECM by producing, degrading and remodeling all of its components. Inversely, ECM components not only participate in tissue organization but also contribute to processes such as cellular maintenance, proliferation, and migration, as well as support for various signaling pathways [2]. The main constituents of the ECM are specific (both in quality and quantity) to each tissue and include structural components as well as a wide variety of enzymes involved in ECM renewal [3]. It is tempting to underline the importance of the participation of proteolytic events in ECM homeostasis, since they imply points of no return in terms of ECM composition. At the same time, a simple in vivo proteolytic analysis is complicated, since multiple regulatory mechanisms that involve different cofactors and inhibitors coexist. Furthermore, situations of nonspecificity and substrate redundancy are common in the proteolytic landscape of any given tissue [4]. Proteolysis not only causes ECM degradation as well as the inhibition of enzyme activity but also, in some cases, the appearance of new activities derived from products generated after the proteolytic event [5]. In any case, the involvement of members of various families of proteinases has been described in a myriad of processes, such as neural development [1,2]. ADAMTSs (*A*
*D*isintegrin *A*nd *M*etalloprotease with *T*hrombo*S*pondin motifs) are a family of proteinases in which some of its members have been described as participating in the degradation of ECM components of the central nervous system (CNS) and thus the regulation of neural physiological or pathological events [6,7].

As occurs in all tissues, the ECM components of the CNS play crucial roles in the organization and maintenance of cellular functions [8]. In fact, the ECM not only forms a scaffold to give support to neural cells but also contributes to stabilizing precise connections and interactions that influence processes such as synaptogenesis, cellular migration and proliferation, and mechanisms of signaling pathways [9,10,11]. These roles are essential for the normal development of the CNS and for repair following traumatic injuries or for repair related to neurogenerative disorders [12]. Again, ECM components and three-dimensional structures are subjected to modification, renewal, and reorganization, in which members of the proteinase family of enzymes participate. In particular, in this review, we will summarize the associations between the ADAMTS family of metalloproteases and normal and pathological situations within the CNS.

## 2. Lecticans in the CNS

Proteoglycans are major constituents of the ECM of the CNS that are involved in cell–cell interactions and cytokine-mediated signaling processes [13,14]. These glycosylated proteins consist of a protein core covalently linked to glycosaminoglycans (GAGs) [15]. GAGs are formed by negatively charged linear polysaccharides that are modified by sulfation and contain a high capacity to trap water, thus allowing the tissues to resist deformation following compressive loads. GAGs are also related to signaling functions, while the protein core is responsible for mediating cell–cell or cell–ECM interactions. Such effects are particularly relevant following a traumatic injury, since the expression of proteoglycans increases considerably in response to damage in the CNS [16]. Astrocytes and neurons are mainly responsible for producing proteoglycans as a mechanism for protecting the damaged region [17,18,19]. In return, this excessive production of proteoglycans inhibits axonal growth and consequently impairs the regeneration process [12]. This inhibitory effect has to be subsequently abrogated through the proteolytic digestion of proteoglycans, which promotes the axonal regrowth process [19,20].

The lectican group, also known as hyalectans, includes some of the main proteoglycans expressed in the CNS [21]. This group comprises versican, aggrecan, brevican, and neurocan, which are molecules characterized by the presence of a central protein core containing attachment sites for GAGs (Figure 1). This long central region is flanked by two globular domains, G1 and G3, which are located at the amino-terminal and the carboxyl-terminal ends, respectively. Aggrecan is the only lectican that contains an additional interglobular domain, G2, which is located in close proximity to G1. Five different isoforms for versican—V0, V1, V2, V3, and V4—have been identified as a result of alternative splicing events that generate regions of different sizes for the attachment of GAGs [22]. Versican V0 is the largest isoform and contains up to 23 GAG attachment sites in two attachment regions, named GAGα and GAGβ. Versican V1 and V2 are isoforms that are shorter than versican V0, with a smaller number of positions for the anchoring of GAGs. In contrast, versican V3 lacks attachment sites for GAGs. Versican V4 is the newest isoform of versican, which has been identified so far in breast cancer and contains a shortened GAGβ region [23]. It is noteworthy that the four lecticans are noncovalently linked to hyaluronic acid through the G1 domain. Hapln proteins stabilize these interactions, preventing the diffusion of lecticans in the ECM. Four members compose the family of Hapln proteins, of which three are found in the CNS (Hapln1, Hapln2, and Hapln4) [24]. Globally, the linked molecules and the interactions with other ECM components contribute to the formation of a three-dimensional network that has an essential role in the maintenance of homeostasis in the CNS [25]. This effect is particularly relevant in those situations in which the expression of lecticans is increased in specific brain areas, such as damaged regions, after traumatic brain injury or in late embryonic and early postnatal mammalian development stages [14,26]. For instance, aggrecan and versican have been found to be expressed in embryos on day 16 in developing rat CNS, in particular in areas of the cerebral cortex, amygdala, or optic and lateral olfactory tracts, among others [27]. Aggrecan is a common component of perineuronal nets, and neurons can influence the differential glycosylation of aggrecan to regulate the microheterogeneity of glycosylation and the organization of perineuronal nets [28]. The absence of the functional gene encoding for aggrecan results in lethality at birth due to major structural abnormalities [29]. However, culture systems derived from cartilage matrix-deficient mice, which lack aggrecan, have contributed to investigating the contribution of aggrecan in the function and composition of perineural nets [30]. The importance of aggrecan in the assembly of perineural nets has also been validated through the employment of an animal model. In fact, the characterization of mice containing a selective deletion in the visual cortex in the gene encoding for aggrecan has revealed the abolition of the perineural net structure [31]. This structural alteration modifies brain plasticity, restoring juvenile plasticity in the visual cortex, and improves the capacity to recognize objects.

With regard to the different isoforms of versican, versican V2, which contains the GAGα region but lacks the GAGβ region, is the isoform expressed predominantly in the nervous system [26,32,33,34]. Moreover, versican V2 can carry out functions other than those performed by the other isoforms. For instance, versican V2 hampers differentiation and activates apoptosis when it is exogenously expressed in PC12 cells, a pheochromocytoma-derived cell line commonly employed to study neural differentiation; however, in stark contrast, versican V1 promotes cell differentiation in this cell line [35,36]. The functional relevance of versican V2 has also been revealed in mice lacking the splice variant of V2 [37]. While the elimination of complete gene coding for full-length versican leads to early embryonic lethality [38], mice lacking isoform V2 are viable and fertile. However, these mice have an ECM that contains important structural aberrations in the nodes of Ranvier [37].

Brevican is the smallest core protein of the lecticans (Figure 1), and its expression begins in late embryonic stages and continues through adulthood [39]. It is produced by oligodendrocytes and astrocytes in white matter, but its expression is differently regulated during development in these two cell types [40]. As happens with aggrecan, brevican is also an important constituent of perineuronal nets, and the impact of this lectican in the spatial coupling of pre- and postsynaptic elements has recently been shown (it thus contributes to precise synaptic transmission in the cochlea) [41]. Likewise, brevican is associated with the axon initial segment through an interaction with neurofascin 186, a glycoprotein belonging to the Ig superfamily [42,43]. Brevican is closely related to neurocan, but the expression patterns of these lecticans differ. Indeed, while brevican is highly expressed in different areas of mature brains, the expression peaks for neurocan can be detected during embryo development, but they decline in adult brains in normal conditions [39,44]. Although both neurocan-deficient mice [44] and brevican-deficient mice [39] are viable and fertile, their phenotypic characterization indicates that these lecticans can display important structural and functional roles in the ECM of some areas of the CNS. Brevican and neurocan double-knockout mice are also viable, without obvious functional deficits [45]. However, the simultaneous absence of brevican and neurocan facilitates the growth of a subpopulation of sensory fibers in the spinal cord dorsal root entry zone (following rhizotomy). A recent study by Gottschling et al. [46] revealed new findings about the roles of brevican and neurocan in the CNS and their functional relationship with other ECM components in the perineuronal nets. These authors characterized quadruple brevican/neurocan/tenascin-C/tenascin-R-deficient mice to demonstrate alterations in their excitatory and inhibitory synaptic responses. In fact, the absence of these four ECM components increased the number of excitatory and reduced the number of inhibitory synaptic molecules. In addition, higher neuronal network activity and the reduction of perineuronal nets in the hippocampus could also be detected in the quadruple knockout mice.

Other proteoglycans expressed in the CNS include phosphacan and receptor-type protein tyrosine phosphatase-β (RPTP-β) [13]. Phosphacan is a secreted proteoglycan generated by the alternative splicing of the gene coding for RPTP-β [47]. Structurally, phosphacan and RPTP-β contain an amino-terminal carbonic anhydrase-like domain, followed by a fibronectin type III domain and by a region with attachment sites for GAGs. Phosphacan lacks the transmembrane domain and the two intracellular tyrosine phosphatase domains identified in the carboxy-terminal region of RPTP-β (Figure 1). Although obvious deficiencies were not initially observed in mice lacking the gene encoding for RPTP-β [48], the presence of phosphacan is essential in developing brains in those areas related to neural cell migration. In this regard, depending on the cellular context, both the promotion of neurite outgrowth in mesencephalic, cortical, and hippocampus neurons of rat embryos [49,50,51] and the inhibition of neurite outgrowth in ganglion cells in the retina [52] have been associated with phosphacan. Moreover, phosphacan has been related to essential roles in the organization of the neural stem cell niche [53].

Finally, reelin is a glycoprotein associated with synaptic plasticity and neurotransmission through the modulation of intracellular components such as Dab1 (Disabled-1) (after an interaction with specific transmembrane lipoprotein receptors such as ApoER2) [54,55]. The contribution of reelin to neuronal embryonic development as well as to adult nervous tissue physiology is demonstrated by the fact that the absence, or proteolytic processing, of this glycoprotein causes important brain abnormalities and might be implicated in neuronal disorders such as schizophrenia and Alzheimer’s disease [54,55,56,57]. Full-length reelin is a 420-kDa glycoprotein that can be proteolytically processed in vivo, which results in functional regulation either through its conversion to an active form or by negative modulation of its activity [56,58,59]. Members of the ADAMTS family of proteases are among the proteases involved in the cleavage of reelin and the concomitant functional implications for brain disorders.

## 3. ADAMTSs in the CNS

The ADAMTSs are a family of secreted proteins that is composed of 19 members in mammals [60,61]. ADAMTSs are involved in a wide variety of physiological and pathological processes that include, among other things, their participation in the degradation and thus the remodeling of components of the extracellular matrix, the inhibition of angiogenesis, and the regulation of inflammatory processes [61,62].

ADAMTSs are secreted enzymes characterized by a complex structure (Figure 2) with different domains that, in general, adjust to the following linear architecture: a prodomain, a metalloprotease domain, a disintegrin domain, a central thrombospondin-1-like domain (TSP), a cysteine-rich region, and a variable number of TSP repeats at the C-terminal end [63]. Characteristic motifs are also present in some domains, for example, a furin recognition sequence at the end of the prodomain, a zinc-binding motif in the metalloproteinase domain with an aspartic residue at the end of the catalytic center, and conserved patterns of cysteine residues in TSP- and cysteine-rich domains. In addition, differences between the family members arise from the presence of specific structural characteristics in terms of particular domains with additional functions for these ADAMTSs. For example, the GON-1 motif is present at the C-terminal end of ADAMTS-9 and ADAMTS-20, and a cubillin (cub) motif can be found only in ADAMTS-13 [63,64,65]. Furthermore, the activity of ADAMTSs can be modulated not only by the presence or absence of certain domains, but also by proteolytic processing through the generation of fragments with new functions or even by protein–protein interactions with other proteins of the ECM [6,66].

ADAMTSs, as part of the family of metalloproteases, can subsequently be classified not only according to their structure but also according to their proteolytic activity toward specific substrates of the ECM. This fact causes ADAMTSs to be associated with processes that occur in specific tissues, depending on the presence or absence of their known substrates. Thus, ADAMTS-1, -4, -5, -8, -9, -15, and -20 are considered to be hyalectanases, since they are able to degrade one or more of the hyalectans that were described in the previous section [67]. Amino-procollagenase activity has been described for ADAMTS-2, -3, and -14, which implies the requirement of these enzymes in the maturation and formation of collagen fibers within the ECM [68]. In cartilaginous tissue, the activity of ADAMTS-7 and -12 has been detected in one of its components, as is the case with the cartilage oligomeric matrix protein (COMP) [69]. Thrombotic thrombocytopenic purpura (TTP) is a rare blood disorder caused by the absence of the specific proteolytic activity of ADAMTS-13 toward the von Willebrand factor, which causes clots in small vessels throughout the whole body [63,70]. On the other hand, some of the ADAMTSs can be considered to be orphan enzymes, since their substrates have not yet been identified and, at present, their classification depends only on similarities in their structure, such as with ADAMTS-6 and -10, ADAMTS-16 and -18, and ADAMTS-17 and -19 [60]. The group of hyalectanases will be the main focus of this review, taking into account the importance of their substrates in physiology as well as their involvement in the mechanisms underlying important neuronal disorders.

Aggrecan can be cleaved by several members of the ADAMTS family of proteinases. However, ADAMTS-4 and ADAMTS-5 can be considered to be the main aggrecanases, since they are more effectively able to degrade this proteoglycan [71,72]. In ADAMTS-4, this activity depends on an interaction between different motifs of both molecules, the central thrombospondin type-1 (TSP-1) motif of ADAMTS-4, and the glycosaminoglycans within the aggrecan structure [73]. Although both ADAMTS-4 and ADAMTS-5 can act upon aggrecan in vitro, the latter was shown to be the major in vivo aggrecanase in mouse cartilage in a mouse model of inflammatory arthritis. Furthermore, ADAMTS-5 also seems to be responsible for the cleavage of this proteoglycan in osteoarthritic patients [74].

Versican has been described as being processed by various ADAMTSs, particularly ADAMTS-1, -4, -5, -9, -15, and -20 [67,75]. Interestingly, the cleavage of versican by ADAMTS-1 generates a 70-kDa bioactive fragment called versikine that is involved in different processes such as apoptosis and the migration of immune cells, thus inhibiting the development of myeloma [5]. In addition, the cleavage of versican by ADAMTSs also plays a very important role in physiological processes such as angiogenesis, ovulation, tissue morphogenesis, and vascular disease [76,77,78].

Brevican is mainly degraded by ADAMTS-1, -4, and -5, generating two possible fragments, 55 and 90 kDa. Both fragments have been associated with the pathological activities of brevican [79,80]. In particular, highly ECM-invasive properties of glioma cells are characterized by ADAMTS-4 overexpression together with a great capacity for brevican cleavage. However, only ADAMTS-5 overexpression has been detected in vivo in human glioma tissue [80]. Moreover, a correlation between an increase in brevican processing by ADAMTS-1 and -4 and the loss of synaptic density has been described [81].

More recently, neurocanase activity has been attributed to ADAMTS-12, and thus it can be considered to be a new hyalectanase [7]. Neurocan degradation by ADAMTS-12 is able to cause changes in the adhesion and migration profiles of the human neuroglioma H4 cell line. The in vivo participation of ADAMTS-12 in neurocan degradation is underlined by the fact that the absence of this protease causes neurocan accumulation in particular areas in the brain of ADAMTS-12-deficient mice [7].

In addition to hyalectans, these ADAMTSs can also degrade other proteoglycans such as phosphacan and reelin [82]. ADAMTS-4 in particular cleaves reelin, blocking its cellular signaling function (its expression and processing are altered during aging) and causing defects in synaptic plasticity and cognitive impairment [83].

The expression patterns of hyalectanases of the ADAMTS family have been detected by different techniques in most CNS structures, including the hippocampus, striatum, cortex, temporal lobe, brain stem, and spinal cord [81,83,84,85,86,87]. In particular, ADAMTS-4 is the most expressed metalloprotease in basal conditions in adult mice [88]. However, ADAMTS-4 mRNA expression increases progressively during the first weeks after birth, and then, in a similar way, its expression decreases in adult mice [37]. In addition, this enzyme has also been detected in postmortem human brains [89]. In vivo and in vitro data have shown that although microglia and neurons also express ADAMTSs, most of them are produced by astrocytes, specifically after a brain injury [90]. In rat brains, ADAMTS-4 has been detected in dentate granular neurons and pyramidal cells [81]. In vitro, the expression of ADAMTS-4 in cortical neurons and cortical microglia has also been described [91,92]. In cultured astrocytes (as mentioned before), ADAMTS-4 is expressed at basal levels, but the presence of tumor necrosis factor-α (TNF-α) stimulates the production of ADAMTS-1 and ADAMTS-4 in these cells [91,93]. ADAMTS-15 is expressed by excitatory thalamic relay neurons in the dorsal thalamus, while in the hippocampus and neocortex, it is generated by inhibitory interneurons [94]. ADAMTS-1 is expressed in mouse and rat brains during development; in motor neurons in injured mice; and in the frontal cortex of humans with Down’s syndrome, Alzheimer’s disease, and Pick’s disease [87,95,96,97]. ADAMTS-9 is expressed in a measured way in the CNS at all stages of mouse development, except in the floor plate of the diencephalon, cerebral cortex, dorsal root ganglia, and choroid plexus [84]. In general, ADAMTSs increase their expression in response to certain diseases, neuronal disorders, or CNS lesions [98]. For example, the expression of ADAMTS-4 increases in pathological situations such as Alzheimer’s disease, ischemic stroke, amyotrophic lateral sclerosis, and spinal cord injury [89,99,100,101]. Inside and outside the CNS, ADAMTSs also have inflammatory and antiangiogenic functions. ADAMTS-1 was the first member of the ADAMTS family to be identified with these properties [52], but other members, such as ADAMTS-12, are also involved in inflammatory processes (mice deficient in this protease have shown a prolonged inflammation phenotype in models of pancreatitis, colitis, and lipopolysaccharide (LPS)-induced inflammation [102].

Consequently, it cannot be ruled out that ADAMTSs contribute to the repair of damaged tissue after brain injuries and also to the progression of neurodegenerative disorders through the convergence of common inflammatory and antiangiogenic properties that have already been assigned to some ADAMTSs. Thus, these proteinases actively participate in all of these processes by regulating the renewal and modification of proteoglycans in the ECM of the CNS [98].

## 4. ADAMTS Functions in Normal and Pathological CNS

As mentioned before, several characteristics make the ECM of the CNS unique in comparison to other organs and tissues of the organism, among them the fact that it contains several proteoglycans, such as reelin, aggrecan, versican, brevican, and neurocan. On the other hand, proteoglycans are responsible for maintaining the integrity of the extracellular matrix of the brain; at the same time, they are the major inhibitors of axon regeneration and plasticity through their presence in glial scar tissue and in perineuronal networks (PNNs) and may affect superior functions such as memory and influence inflammatory reactions after brain injury [103].

Due to their proven involvement in normal and pathological processes, not only in developed tissue but also during the different stages of brain development [21], the existence of finely tuned mechanisms controlling their production, modification, and replacement is essential. In this sense, several studies have described their involvement in the events of proteolytic mechanisms mediated by metalloproteases of the ADAMTS family [94], in particular after CNS damage [104]. Specifically, their participation is important in the degradation of glial scarring and in the posterior stimulation of axonal growth, thus increasing the neuronal synaptic plasticity induced after brain injury [98]. However, the role of ADAMTSs in chronic diseases of the central nervous system is complex and has not been sufficiently explored. For that reason, one of the current challenges is to unravel their specific role in normal and pathological processes in the CNS. For example, in the case of a brain injury derived from a traumatic injury (TBI) or from cerebral ischemia, a cascade of signals occurs that causes the elevation or migration of different components of the ECM to the injured area. The participation of ADAMTS-1, -4, -5, -9, -12, and -13 has been described in this repair process [7,100,105].

ADAMTS-13 has been widely studied for its ability to degrade the von Willebrand factor (vWF), a high-molecular-weight proteoglycan that participates in platelet aggregation by establishing interactions between platelet surfaces and vascular wall components [106]. ADAMTS-13 is responsible for degrading the multimeric vWF chains, contributing to proper homeostasis in thrombus formation [107]. As mentioned earlier, the loss of its function leads to the accumulation of the von Willebrand factor and causes thrombotic thrombocytopenic purpura (TTP) [108]. Analogously, and in the context of brain injury, the vWF plays an important role in hemostasis by recruiting platelets at the site of vascular injury. It is stored in Weibel–Palade bodies in endothelial cells and in platelet granules and is released into circulation after trauma [105]. This release is mediated by ADAMTS-13, so that ADAMTS-13 deficiency is associated with occlusive diseases such as myocardial infarction and stroke, and its low activity is a predictor of unfavorable results in patients with ischemic stroke undergoing endovascular therapy [109].

In mice, recent studies have demonstrated that ADAMTS-13 proteolytic activity is able to exert a protective role during strokes. This function seems to be relevant in mice with fluid percussion injuries, since the administration of recombinant ADAMTS-13, both before and after injury, reduces the reactivity of the vWF, protects the integrity of endothelial cell barriers, and prevents TBI-induced coagulopathy. Recombinant ADAMTS-13 acts by enhancing vWF elimination, but does not affect basal hemostasis [110]. Similarly, inflammatory responses provoked by induced intracerebral hemorrhaging can be limited through the administration of recombinant ADAMTS-13 and a concomitant reduction of vWF activity. In one study, intracerebral hemorrhaging was induced in mice through an intracerebral blood infusion, and after the administration of the recombinant protein, a reduction in inflammatory mediators (such as IL-6), inflammatory cytokines, myeloperoxidase activity, microglial activity, and neutrophil recruitment was observed. Therefore, the treatment of mice with recombinant ADAMTS-13 reduces cerebral edema and, at the same time, the volume of the hemorrhagic lesion [111]. In a similar set of experiments, the administration of a gain-of-function variant of ADAMTS-13 (GoF ADAMTS-13) showed a protective effect in mice that had a cerebrovascular injury induced through occlusion of the middle cerebral artery [112].

It is not just the regulation of hemostasis that causes effects on the ECM of the CNS, since other members of the ADAMTS family of proteoglycanases, such as ADAMTS-1, -4, -5, -9, and -12, participate in its regulation and modification and have important functions in processes such as neuroplasticity, inflammation, and repair through the degradation of proteoglycans that may prevent axial growth or wound closure [7,90]. Thus, an increase in the expression of ADAMTS-1, -4, -5, and -9 has been detected in isolated astrocytes from postnatal zero-day mouse brains in the presence of inflammatory cytokines such as IL-1 [90]. Inflammatory-responsible elements have been detected and characterized in the ADAMTS-9 gene promoter using chondrocytes and chondrosarcoma cells, which caused an increase in ADAMTS-9 levels in the presence of proinflammatory cytokines such as IL-1β [113]. These elements might also be responsible for the elevated levels of ADAMTS-9 (mRNA and proteins) that occur after brain injury provoked by the occlusion of the middle cerebral artery (tMCAo), a known model of focal cerebral ischemia in rats. Through in situ hybridization, the authors of one study showed that ADAMTS-9 expression was confined to neurons of the damaged tissue [114]. In addition, the contribution of ADAMTS-1 and -4 to resolving the experimental stroke elicited after tMCAo might be an important step in enabling the infiltration of inflammatory cells that contribute to brain injury and posterior resolution [93].

In all of the above situations, it seems that alterations in ADAMTS expression are somehow related to the development of inflammatory processes, e.g., TBI or ischemia, which, over time, is consistent with the generation of CNS damage. In the event of an injury, proteoglycan expression increases in the damaged area in order to promote the repair of the lesion. However, their expression has a dual role, since they are able to promote as well as inhibit neuronal growth depending on modifications of the ECM, which in turn serves to sustain cell formation, mobility, and growth factor and cytokine interactions [115]. Therefore, to maintain cellular homeostasis, this newly formed ECM, which is needed to support and sustain the repair process, has to be eliminated once it has fulfilled its scaffolding role in order to eliminate an environment that is nonpermissive of axonal regeneration in the glial scar. ADAMTS-1, -4, -5, -9, and -15 are the main ADAMTSs in the brain known to degrade different proteoglycans, as well as reelin [82,90]. In fact, inflammation markers such as IL-1 are able to induce ADAMTS-1, -4, -5, and -9, and these events correlate in time with an increase in proteoglycan degradation during the early phases of injury progression [116,117].

As mentioned earlier, ADAMTS-4 is the most frequently observed hyalectanase that is expressed at basal levels in the adult brain; at the same time, it is also the most relevant member of the ADAMTS family in terms of neuroreparation after CNS lesions. The in vitro efficiency of ADAMTS-4 in degrading proteoglycans has been widely demonstrated [118]. Furthermore, this proteolytic activity toward proteoglycans seems to not be necessary to stimulate neurite extension in cultured neurons, but at the same time, proteolysis creates a more favorable matrix environment for neurite outgrowth [119]. The positive participation of ADAMTS-4 in enhancing neuroplasticity has also been described after the administration of tissue plasminogen activator (tPA) in a model of spinal cord injury in rats (compression-induced). There, tPA administration stimulated proteoglycan elimination through ADAMTS-4 activation, which contributed to axonal regeneration, sprouting, and functional recovery of the injured area [101]. The direct administration of ADAMTS-4 in rats with an induced spinal cord bruise injury was also able to restore motor function by enhancing axonal regeneration after the injury [92]. In another study, ADAMTS-4- and ADAMTS-5-deficient mice accumulated versikine (but not specific fragments derived from brevican or aggrecan proteolysis) after a spinal cord injury, which suggests that versican is the preferred mediator of both ADAMTSs in neuronal function regeneration [120]. In yet another study, ADAMTS-4-deficient mice showed a motor deficit that seemed to derive from abnormal myelination and electrical nerve activity in adult mice. In fact, ADAMTS-4 is expressed in wild-type animals in oligodendrocytes, which are the cells responsible for myelination in the CNS [121]. Therefore, the participation of ADAMTS-4 in axonal growth either in vivo or in vitro may depend in part on its contribution to myelination processes under normal conditions and thus the regulation of motor capacities in adult mice [122].

Recently, it has been suggested that ADAMTS-12, a known enzyme involved in inflammatory processes [102,123], might also participate in CNS repair processes through the elimination of neurocan [7]. It is known that ADAMTS-12 accumulates in areas of inflammation, and at the same time, both neurocan expression and ADAMTS-12 expression are more evident during embryonic phases. This fact, together with neurocan accumulation in specific areas of the CNS in ADAMTS-12-deficient mice, suggests that ADATMTS-12 neurocanase activity is responsible for the elimination of neurocan in affected tissues either during normal development or during repair processes [7,102]. Neurocan degradation by ADAMTS-12 produces a 50-kDa specific band that resembles those observed after the digestion of versican, brevican, and aggrecan by other ADAMTSs (versikine, for example). The evidence in the literature seems to support this link between neurocan and ADAMTS-12, since they are also associated with certain brain disorders such as schizophrenia and bipolar disorder [124,125,126,127] (at least in the case of ADAMTS-12, also with narcolepsy) [128]. It is not just ADAMTS-12 that can be linked to neuropathies, since ADAMTS-1 (as an example) is overexpressed in the frontal cortex of brains of patients with Down’s syndrome, Alzheimer’s disease (AD), and Pick’s disease, and its presence has been suggested as being a good marker of neurodegeneration [87].

It is known that reelin is a secreted signaling glycoprotein that is largely expressed in the brain: it is crucial to development, both during embryonic and postnatal periods. Reelin is also a key in the regulation of brain functions and synaptic functions in adulthood, and it seems to be mandatory for neural superior functions such as learning and memory [54,55,64,129]. Reelin interacts with several cellular receptors, and a lack of reelin interaction is associated with the appearance of neuropsychiatric diseases such as AD and schizophrenia. Therefore, proteolytic activity against reelin is important for maintaining brain function. The elimination of reelin has been ascribed mainly to members of the ADAMTS family of proteases, more specifically to ADAMTS-2 and ADAMTS-3 [130]. In fact, both ADAMTSs have been involved in both pathologies because both proteases are capable of performing a specific proteolytic cleavage of reelin (N-t cleavage) that eliminates its biological activity. In addition, studies on ADAMTS-2-deficient and ADAMTS-3-deficient mice have shown that this N-t cleavage of reelin diminishes in an important way, specifically in the postnatal cerebral cortex and hippocampus [130,131,132]. The fact that reelin is able to antagonize the deposition and toxicity of beta amyloid (Aβ) peptides in AD suggests the possibility of using inhibitors of ADAMTS-3 proteolytic activity to block reelin N-t cleavage. One study targeted this proteolytic activity by crossbreeding drug-inducible ADAMTS-3-deficient mice with a “next-generation” Alzheimer’s model, and it proved reelin is a putative treatment for this neurological disease [131]. Analogously, *DISC1* (*Disrupted in Schizophrenia 1*) is a known gene that codes for a structural protein that is important in the developing cortex and that is involved in mental illness pathologies such as schizophrenia. DISC1 acts upstream of reelin in the perinatal cerebral cortex and regulates its activity through ADAMTS-4-dependent proteolytic cleavage [56,133].

The processing of reelin by ADAMTS-4 and its implication in neuronal disorders has been described more deeply in terms of AD. For example, the treatment of primary cultures of astrocytes with deposits of Aβ peptides clearly induces ADAMTS-4 transcription [99]. In addition, in a model of transgenic AD mice, tPA was proven to activate both ADAMTS-4 and ADAMTS-5 proteolytic processing of reelin (with expression patterns overlapping in the hippocampus) [83,101]. Moreover, the levels of ADAMTS-5 and tPA increased in AD transgenic mice, while during normal aging, no significant changes were detected in the levels of these proteases or in the processing of reelin [83]. Finally, a recent study found a large fraction of insoluble Aβ peptides truncated at the N-terminus with Aβ4-x peptides in the brains of Alzheimer’s patients (autopsies): this processing is carried out by ADAMTS-4. High levels of Aβ4-x peptides have been observed in animals deficient in ADAMTS-4 in an 5xFAD mice model, which was used as an amyloidosis model for the study of the accumulation of this peptide [134].

## 5. Concluding Remarks

The ECM composition of the CNS includes a myriad of components with very different natures that affect all aspects of tissue development and function. Within the CNS, proteoglycans are known to participate in cell–cell interactions and also in several signaling events. Therefore, the regulation of the synthesis, modification, and degradation of these ECM components is of crucial importance in physiological and pathological events of the CNS. In this review, we have tried to summarize how the degradation of CNS proteoglycans by members of the ADAMTS family of proteinases affects functions of the CNS, such as neuroplasticity, tissue repair, and neurological disorders (Table 1). The studies described herein illustrate the most relevant examples of the importance of proteoglycan degradation to the normal development and functioning of the CNS. In addition, the ECM environment should be considered to be a complex ecosystem in which proteolytic events elicited by ADAMTSs can also be modified by other components of the ECM [6]. A deep knowledge of the biology of the components of the ECM of the brain would help offer more achievable therapeutic approaches to enhancing repair mechanisms or even reducing pain episodes caused by injuries or illness. In this regard, the characterization of the enzymatic degradation of these components in both normal and pathological conditions may provide future therapeutic strategies for treating brain disorders [26,45].

## Figures and Tables

**Figure 1 biomolecules-10-00403-f001:**
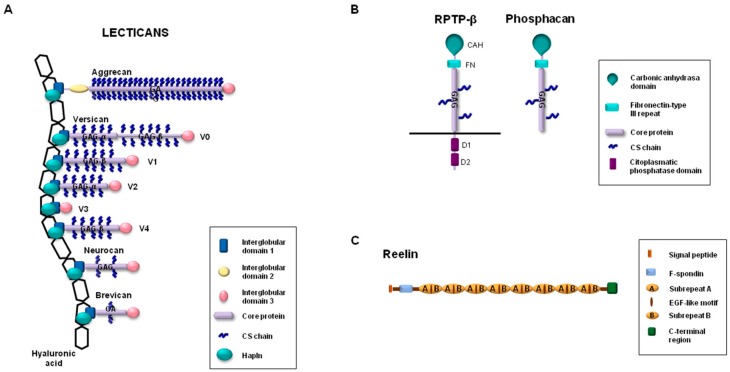
Schematic representation of (**A**) lecticans, (**B**) RPTP-β (receptor-type protein tyrosine phosphatase-β), phosphacan, and (**C**) reelin.

**Figure 2 biomolecules-10-00403-f002:**
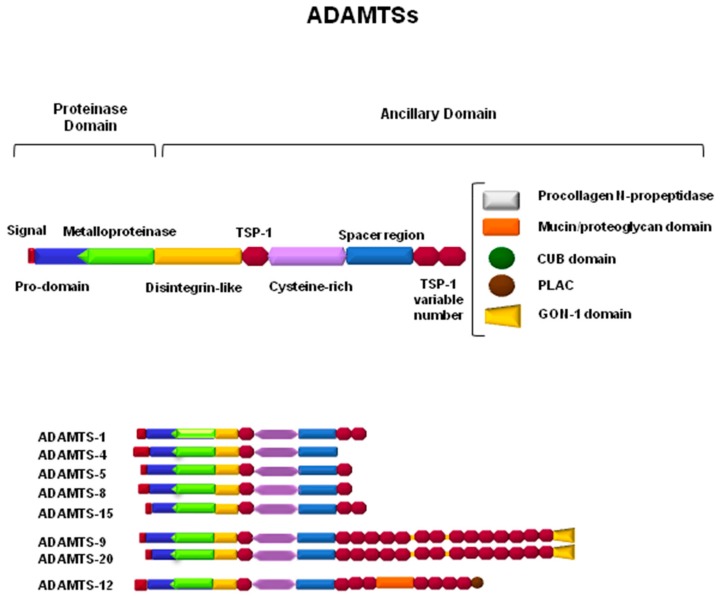
Schematic representation of ADAMTSs involved in the proteolytic processing of components of the central nervous system extracellular matrix (ECM).

**Table 1 biomolecules-10-00403-t001:** ADAMTSs in the central nervous system (CNS).

ADAMTS	Known Substrates	Neuronal Process/Disorder
ADAMTS-1	Versican; brevican	Stroke [93]; spinal cord injury [117]; neuroplasticity [104]; inflammation [93,116,117]; Down’s syndrome [87]; Alzheimer’s disease [87]
ADAMTS-3	Reelin	Alzheimer’s disease [130,131]; schizophrenia [130]
ADAMTS-4	Versican; aggrecan; reelin; brevican	Stroke [93]; spinal cord injury [117]; neuroplasticity [83,119]; inflammation [93,116,117]; myelination [121]; Alzheimer’s disease [83,134]; schizophrenia [56]
ADAMTS-5	Versican; aggrecan; reelin; brevican	Stroke [93]; spinal cord injury [117]; neuroplasticity [83]; inflammation [116,117]; Alzheimer’s disease [83]
ADAMTS-9	Versican	Stroke [114]; spinal cord injury [117]; inflammation [113,116,117]
ADAMTS-12	Neurocan	Inflammation [7,123]; schizophrenia [7,124,127]
ADAMTS-13	von Willebrand factor	Inflammation [111]; stroke [109,110,112]

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
