# Peer review of "New Insights into ADAMTS Metalloproteases in the Central Nervous System"

_biomolecules, 2020, doi:10.3390/biom10030403_

Round 1

Reviewer 1 Report

This review is well organized and it comprehensively includes the role of proteoglycan as well as ADAMTS.

Here are my suggestions.

1) Although the importance of hyalectans, they did not mention about the Hapln family (hyaluronan and proteoglycan binding link protein gene family). Authors should add more information in this regard.

2) In the Page 5, line 215; Ref #68 should not be cited or by mistake. It is a report of ADAMTS5 KO mice with OA model. They don’t give any information regarding the role of ADAMTS5 in human OA patient.

3) The role of ADAMTS-1, 4, and 5 in brevican degradation, there are several papers and thus not conclusive. That is, not particular ADAMTS member is critical at the current moment and they should be thoughtful in this regard.

4) Authors should cite the recent review in Matrix Biology last year (doi: 10.1016/j.matbio.2018.06.002.)

Minor:

1) Who is the last author because the list is not completed.

2) In the Abstract, ADAMS are family of “ A disintegrin…” not “disintegrin….” .

3) Fig. 2 should be revised because it is too small.

Author Response

Answers are given after each point of the reviewer

This review is well organized and it comprehensively includes the role of proteoglycan as well as ADAMTS.

Here are my suggestions.

1) Although the importance of hyalectans, they did not mention about the Hapln family (hyaluronan and proteoglycan binding link protein gene family). Authors should add more information in this regard.

We have introduced a sentence mentioning the Hapln family of proteins in stabilizing hyaluronic acid and lecticans interaction. Furthermore, in Figure 1, we have added schematically the position of Hapln proteins.

2) In the Page 5, line 215; Ref #68 should not be cited or by mistake. It is a report of ADAMTS5 KO mice with OA model. They don’t give any information regarding the role of ADAMTS5 in human OA patient.

This reference has been taken away since it referred to a mice model. We have included a new reference abotu the role of ADAMTS-4 and ADAMTS-5 in huma OA

3) The role of ADAMTS-1, 4, and 5 in brevican degradation, there are several papers and thus not conclusive. That is, not particular ADAMTS member is critical at the current moment and they should be thoughtful in this regard.

We don´t quite understand what the reviewer want us to modify about their participation in brevican degradation. We have found several papers describing brevican degradation by these ADAMTSs so we think we can keep our review the way it is:

Human glioblastomas overexpress ADAMTS-5 that degrades brevican. Nakada M, Miyamori H, Kita D, Takahashi T, Yamashita J, Sato H, Miura R, Yamaguchi Y, Okada Y. Acta Neuropathol. 2005 Sep;110(3):239-46. Epub 2005 Aug 30.

ADAMTS4 and ADAMTS5 knockout mice are protected from versican but not aggrecan or brevican proteolysis during spinal cord injury. Demircan K, Topcu V, Takigawa T, Akyol S, Yonezawa T, Ozturk G, Ugurcu V, Hasgul R, Yigitoglu MR, Akyol O, McCulloch DR, Hirohata S. Biomed Res Int. 2014;2014:693746. doi: 10.1155/2014/693746. Epub 2014 Jul 3.

Association between protease-specific proteolytic cleavage of brevican and synaptic loss in the dentate gyrus of kainate-treated rats. Yuan W, Matthews RT, Sandy JD, Gottschall PE. Neuroscience. 2002;114(4):1091-101.

Brevican is degraded by matrix metalloproteinases and aggrecanase-1 (ADAMTS4) at different sites. Nakamura H, Fujii Y, Inoki I, Sugimoto K, Tanzawa K, Matsuki H, Miura R, Yamaguchi Y, Okada Y. J Biol Chem. 2000 Dec 8;275(49):38885-90.

Brain-enriched hyaluronan binding (BEHAB)/brevican cleavage in a glioma cell line is mediated by a disintegrin and metalloproteinase with thrombospondin motifs (ADAMTS) family member. Matthews RT, Gary SC, Zerillo C, Pratta M, Solomon K, Arner EC, Hockfield S. J Biol Chem. 2000 Jul 28;275(30):22695-703.

The endogenous proteoglycan-degrading enzyme ADAMTS-4 promotes functional recovery after spinal cord injury. Tauchi R, Imagama S, Natori T, Ohgomori T, Muramoto A, Shinjo R, Matsuyama Y, Ishiguro N, Kadomatsu K. J Neuroinflammation. 2012 Mar 15;9:53. doi: 10.1186/1742-2094-9-53.

Matrix-degrading proteases ADAMTS4 and ADAMTS5 (disintegrins and metalloproteinases with thrombospondin motifs 4 and 5) are expressed in human glioblastomas. Held-Feindt J, Paredes EB, Blömer U, Seidenbecher C, Stark AM, Mehdorn HM, Mentlein R. Int J Cancer. 2006 Jan 1;118(1):55-61.

Altered production and proteolytic processing of brevican by transforming growth factor beta in cultured astrocytes. Hamel MG, Mayer J, Gottschall PE. J Neurochem. 2005 Jun;93(6):1533-41.

4) Authors should cite the recent review in Matrix Biology last year (doi: 10.1016/j.matbio.2018.06.002.)

This suggested refence has been cited

Minor:

1) Who is the last author because the list is not completed.

Fixed

2) In the Abstract, ADAMS are family of “ A disintegrin…” not “disintegrin….” .

Fixed

3) Fig. 2 should be revised because it is too small.

We enlarge the figure in our manuscript. If needed to be we can send the figure in other format

Reviewer 2 Report

This is an interesting review on the role of ADAMTSs in the CNS. There are a few point that needs to be addressed:

1-There a few typos and punctuations that need to be corrected:

For example

Page 2, line 58. Space after [9-11]. These

Page 2, line 72: should be spelled mainly

Page 7, line 278: there is a word missing “It is necessary the existence”? That the existence? And what is a “tuned” mechanism? This is hard to understand the biological meaning of a tuned mechanism? Maybe give an example?

Page 7, 280: there is a missing word “those events a multitude”

2- In the Figure 1 legend, write in full this protein RPTPbeta: I write in full to avoid confusion. Is it the Ptprz1 gene according to uniprot.org? There are multiple terms that could lead to confusion and it could help to clarify.

3- page 4, line 164-165, what does “functional implications of reelin” mean? This sentence is confusing and should be re-written. Do the authors imply that processing of reelin lead to changes in biological functions?

4- page 4, 167 ADAMTSs was already described earlier in the manuscript. It can be removed.

5- I do not agree with the title in Table 1 “Main substrates.” There are several unknown substrates about ADAMTSs and very limited publications have investigated these proteases in context of diseases or in vivo conditions. I would suggest changing it to “known substrates”.

Author Response

Answers are given after each reviewers point

This is an interesting review on the role of ADAMTSs in the CNS. There are a few point that needs to be addressed:

1-There a few typos and punctuations that need to be corrected:

For example

Page 2, line 58. Space after [9-11]. These

Fixed

Page 2, line 72: should be spelled mainly

Fixed

Page 7, line 278: there is a word missing “It is necessary the existence”? That the existence? And what is a “tuned” mechanism? This is hard to understand the biological meaning of a tuned mechanism? Maybe give an example?

Page 7, 280: there is a missing word “those events a multitude”

We have tried to make the sentence clearer. Hopefully the English Editing by MDPI improves even more the meaning of the sentence

2- In the Figure 1 legend, write in full this protein RPTPbeta: I write in full to avoid confusion. Is it the Ptprz1 gene according to uniprot.org? There are multiple terms that could lead to confusion and it could help to clarify.

In the legend for the figure is written RPTPbeta (receptor-type protein-tyrosine phosphatase beta).

3- page 4, line 164-165, what does “functional implications of reelin” mean? This sentence is confusing and should be re-written. Do the authors imply that processing of reelin lead to changes in biological functions?

The sentence has been rewritten to make it clearer. Hopefully the English Editing by MDPI improves even more the meaning of the sentence.

4- page 4, 167 ADAMTSs was already described earlier in the manuscript. It can be removed.

The acronim was eliminated since it is already in the abstract

5- I do not agree with the title in Table 1 “Main substrates.” There are several unknown substrates about ADAMTSs and very limited publications have investigated these proteases in context of diseases or in vivo conditions. I would suggest changing it to “known substrates”.

We agree with the reviewer and made the change.

Reviewer 3 Report

This is a useful overview of an area of emerging importance. I have a few minor text changes:

line 4 - delete the final "and"

line 17 - "In the central...."

line 26 -"may suggest their..."

line 41 - "It is tempting...."

line 72 - "neurons are mainly..."

line 77 - "some of the main..."

line 83 -"....and V4 have been identified as a result of..."

line 97- delete "precisely the"

Figure 1 and 2 - both figures are rather small to see details, and this is also the case for the keys in Figure 1

line 144 - "are phosphacan and the...."

line 150 - "Although obvious deficiencies were not initially..."

line 150 - "the presence of..."

line 155 - "of the neural stem cell niche"

line 159/160- "The contribution of Reelin to neuronal....is demonstrated by the fact that the absence...."

line 162 - "neuronal disorders"

line 181 - "GON-1 motif"

line 184 - "through their proteolytic..."

line 200- "On the other..."

Line211/212- "Although both ADAMTS-4 and ADAMTS-5 can act...., the latter has been..."

line 243- "in adult mice"

line 248 - "as mentioned before"

line 278-279 - "[21], the existence of tuned.....and replacement is essential"

line 287 - "their specific roles"

line 312 - myeloperoxidase

line 337- "However, their expression..."

Line 347 - "As mentioned earlier........is the most frequently observed hyalectanase..."

line 349 - ADAMTS-4

line 368 - "might also participate"

line 416 - The ECM composition..."

line 417 "Within the CNS..."

Author Response

Answers are given after each reviewers point

This is a useful overview of an area of emerging importance. I have a few minor text changes:

line 4 - delete the final "and"

Fixed

line 17 - "In the central...."

Fixed

line 26 -"may suggest their..."

Fixed

line 41 - "It is tempting...."

Fixed

line 72 - "neurons are mainly..."

Fixed

line 77 - "some of the main..."

Fixed

line 83 -"....and V4 have been identified as a result of..."

Fixed

line 97- delete "precisely the"

Fixed

Figure 1 and 2 - both figures are rather small to see details, and this is also the case for the keys in Figure 1

We enlarge the figures in the manuscript. Further enlargement can be made by the editoiral. We also can upload de figures in other format if needed to be.

line 144 - "are phosphacan and the...."

Fixed

line 150 - "Although obvious deficiencies were not initially..."

Fixed

line 150 - "the presence of..."

Fixed

line 155 - "of the neural stem cell niche"

Fixed

line 159/160- "The contribution of Reelin to neuronal....is demonstrated by the fact that the absence...."

Fixed

line 162 - "neuronal disorders"

Fixed

line 181 - "GON-1 motif"

Fixed

line 184 - "through their proteolytic..."

Fixed

line 200- "On the other..."

Fixed

Line211/212- "Although both ADAMTS-4 and ADAMTS-5 can act...., the latter has been..."

Fixed

line 243- "in adult mice"

Fixed

line 248 - "as mentioned before"

Fixed

line 278-279 - "[21], the existence of tuned.....and replacement is essential"

Fixed

line 287 - "their specific roles"

Fixed

line 312 - myeloperoxidase

Fixed

line 337- "However, their expression..."

Fixed

Line 347 - "As mentioned earlier........is the most frequently observed hyalectanase..."

Fixed

line 349 - ADAMTS-4

Fixed

line 368 - "might also participate"

Fixed

line 416 - The ECM composition..."

Fixed

line 417 "Within the CNS..."

Fixed